# Distinct Food Triggers for Migraine, Medication Overuse Headache and Irritable Bowel Syndrome

**DOI:** 10.3390/jcm12206488

**Published:** 2023-10-12

**Authors:** Merve Ceren Akgor, Doga Vuralli, Damla Hazal Sucu, Saliha Gokce, Bahar Tasdelen, Fatih Gultekin, Hayrunnisa Bolay

**Affiliations:** 1Department of Neurology and Algology, Faculty of Medicine, Gazi University, Ankara 06560, Türkiye; cerenmervee@gmail.com (M.C.A.); dogavuralli@hotmail.com (D.V.); salihagokce97@gmail.com (S.G.); 2Neuroscience and Neurotechnology Center of Excellence (NÖROM), Gazi University, Ankara 06560, Türkiye; 3Neuropsychiatry Center, Gazi University, Ankara 06560, Türkiye; 4Department of Biostatistics and Medical Informatics, Faculty of Medicine, Mersin University, Mersin 33343, Türkiye; dmlhzl5@gmail.com (D.H.S.); bahartasdelen@gmail.com (B.T.); 5Department of Medical Biochemistry, Lokman Hekim University, Ankara 06510, Türkiye; drfatih2000@gmail.com

**Keywords:** migraine, medication overuse headache, irritable bowel syndrome, food triggers, cognitive impairment, dopaminergic food, food additives, histamine

## Abstract

Background: Irritable bowel syndrome (IBS) is an under-diagnosed common health problem that impairs quality of life. Migraine and IBS are comorbid disorders that are triggered by foods. We aim to investigate IBS frequency in medication overuse headache (MOH) patients and identify food triggers and food avoidance behavior. Methods: Participants who completed the cross-sectional, observational and online survey were included (*n* = 1118). Demographic data, comorbid disorders, medications used, presence of headache, the diagnostic features of headache and IBS, migraine related subjective cognitive symptoms scale (MigSCog), consumption behavior of patients regarding 125 food/food additives and food triggers were asked about in the questionnaire. Results: Migraine and MOH diagnoses were made in 88% and 30.7% of the participants, respectively. Non-steroidal anti-inflammatory drugs (NSAIDs) were the main overused drug (89%) in MOH patients. IBS symptoms were present in 35.8% of non-headache sufferers, 52% of migraine patients and 65% of MOH patients. Specific food triggers for MOH patients were dopaminergic and frequently consumed as healthy foods such as banana, apple, cherry, apricot, watermelon, olive, ice cream and yogurt. MigSCog scores were significantly higher in episodic migraine and MOH patients when IBS symptoms coexisted. Conclusions: The frequency of IBS was higher in MOH patients compared to migraine patients. Coexistence of IBS seems to be a confounding factor for cognitive functions. MOH specific triggers were mostly dopaminergic foods, whereas migraine specific food triggers were mostly histaminergic and processed foods. Personalized diets focusing on food triggers and interference with leaky gut must be integrated to MOH and migraine treatment to achieve sustainable management of these disorders.

## 1. Introduction

Medication overuse headache (MOH) is a chronic secondary headache that causes serious disability and reduced productivity. Approximately 2% of the population is affected by medication overuse headache [1]. MOH mainly develops in patients with primary headaches, especially migraines, due to frequent use of analgesic medications. 

Migraine is often accompanied by various comorbidities such as psychiatric disorders including depression and anxiety, fibromyalgia, sleep disorders, irritable bowel syndrome (IBS) and subjective cognitive complaints. Studies have indicated a correlation between migraine patients and the presence of symptoms related to IBS, suggesting an association between these two conditions. Additionally, there has been long standing recognition that both diseases can be provoked by certain food items. However, in the case of MOH, there is currently no existing research regarding IBS and food triggers.

Several stimuli can trigger a migraine headache in susceptible individuals. Approximately 30% of migraine patients have reported that certain foods trigger their migraine attacks; chocolate, aged cheeses, milk and dairy products and alcohol are well-known examples [2]. Even though MOH frequently accompanies chronic migraine, it remains to be elucidated they have common food triggers.

IBS is a gastrointestinal disorder characterized by chronic, recurrent abdominal pain and variable bowel habits. Depression, anxiety, somatic disorders and fibromyalgia are commonly associated comorbidities with IBS. The quality of life of individuals with IBS is negatively impacted due to the symptoms of the disease and frequent presence of comorbid conditions [3]. Approximately 40% of individuals who meet the diagnostic criteria for IBS do not have official diagnoses [4].

The gut and the brain interact bidirectionally and many individuals with migraines also experience comorbid conditions such as IBS and celiac disease. Migraine and IBS share several common features, including a higher prevalence in women, a chronic and recurrent nature of pain, similar triggering factors, the presence of central sensitization and the occurrence of similar comorbid conditions [5]. Migraine and IBS are both common health problems that impair quality of life and have similar comorbidities. They are both associated with increased healthcare costs and absenteeism in the workplace [6]. Studies conducted in recent years have shown that migraine patients are more likely to have IBS symptoms and that there is a link between these two diseases. Moreover, it has been known for many years that both diseases are triggered by foods. 

In this cross-sectional, observational and online survey study, we hypothesized that MOH patients have higher frequency of IBS symptoms, higher disability rate and different food triggers compared to those without MOH. We aimed to investigate the presence of IBS symptoms in migraine and MOH patients and the food triggers of migraine, MOH and irritable bowel syndrome and the consumption behavior of 125 food and food additive triggers.

## 2. Methods

### 2.1. Study Design and Data Collection

This study was designed as a cross sectional and observational online survey. This prospective study was conducted between September and December 2022. The participants were informed about the survey via social media. The participants filled the questionnaire voluntarily and the introduction section of the survey included an online informed consent form. After obtaining consent, participants were directed to the survey questions. The surveys were filled out anonymously by voluntary individuals and multiple submissions by the same person were electronically prevented. The data were stored electronically only accessible to the researchers.

A survey comprising of four distinct sections has been developed utilizing the platform of Google Forms. The first section includes questions about the age, gender, weight, height, existing medical conditions, medication use and foods that are avoided since childhood. The second section investigates whether participants experience headaches and if so, the characteristics of the pain and associated symptoms (frequency, duration and intensity, presence of photophobia, phonophobia, osmophobia, aggravation of pain with movement, nausea, vomiting), presence of aura/self-administered visual aura rating scale questionnaire [7] and the use of analgesics that fulfill the diagnostic criteria for MOH(International Classification of Headache Disorders-ICHD-3) [8] and the presence of diagnostic IBS symptoms according to Rome IV Criteria [9]. All headaches were diagnosed according to ICHD-3 [8]. For female participants, the survey explicitly excluded the days of menstruation when assessing abdominal pain complaints. In the third section, participants were asked about their consumption behavior regarding a total of 125 food items and food additives. The frequency of consumption was categorized as ‘never’, ‘occasionally’ or ‘frequently’. Additionally, participants were asked whether these 125 food items and additives triggered headaches or abdominal pain and whether they specially triggered migraine attacks, abdominal pain or had no effect on their symptoms. Some of these food items included pastries, simit (Turkish bagel), bread, flatbread, cream cheese, clotted cream, cream, milk, yogurt, butter and butter-based products, sausage, cured beef, salami, hot dogs, smoked meats, olives, orange juice, apples, apricots, bananas, strawberries, red grapes, watermelon, oranges, tangerines, grapefruits, pomegranates, cherries, dried fruits (apricots, grapes, plums, figs, etc.), tomatoes, tomato juice, bell peppers, beans, red carrots, beets, onions, garlic, eggplants, spinach, cauliflower, broccoli, cabbage, mushrooms, dried vegetables (tomatoes, peppers, eggplants, etc.), lentils, chickpeas, kidney beans, hamburgers, pizza, French fries, other fried foods, pasta, bulgur wheat and other wheat derivatives such as semolina and couscous, red meat, processed meat, chicken, fish, shellfish, canned tuna, canned foods, pickles, vinegar, packaged instant noodles, ready-made soups, ketchup, mayonnaise, mustard, ready- made sauces (curry, creamy mushrooms, pesto, etc.), soy sauce, ready-to-eat frozen meat and chicken products, meat/chicken bouillon, seasoned and spicy nuts, chips, tahini, peanut butter, pistachios, hazelnut, sunflower seeds, artificially sweetened snacks, chewing gum, sweet sauces, coffee flavors, foods containing gelatin, biscuits, cakes, bars, crackers, red-colored candies, chocolates and cocoa-containing products, baklava and other syrup-based desserts, ice cream, cream-filled pastries, coffee cream, milk powder, packaged fruit juices, soda, beer, raki, red wine and white wine. Additionally, this section examined whether consuming high amounts of carbohydrates or overeating in a single meal triggered headaches and the speed of eating (fast or slow) was asked. Lastly, migraine related subjective cognitive complaints were assessed by migraine related subjective cognitive symptoms scale (MigSCog) questions [10].

In conclusion, 1191 participants filled out the questionnaire (Figure 1). In total, 43 participants who did not complete the questionnaire and 2 participants who were under the age of 18 and 5 participants who were over the age of 65 were excluded from the study. Additionally, 3 participants with inflammatory bowel disease, 3 participants with cancer and 17 participants with gastric ulcer or reflux were excluded from the survey. The remaining 1118 participants that completed the online survey were included in the study (Figure 1). 

### 2.2. Statistical Analysis

The analyses were performed using ‘Statistica v.13.3.1’ software. Descriptive statistics for numerical variables are given as Mean ± Standard Deviation, while categorical variables are presented in the table as counts (n) and percentages (%). An independent samples *t*-test was conducted for binary group comparisons in this study. Categorical variables were analyzed using Pearsons’s Chi-Square Test. A significance level of *p* < 0.05 was considered statistically significant. The Mig-Scog graphics were created by using the JASP software version 0.14.

In addition, association rule mining (ARM) was used to detect the associations of food triggers in patients with migraine and MOH and to determine how often these similarities occur together with IBS. ARM is employed to determine complex relationships between categorical variables and discover patterns in big data [11]. The ARM method defines the relationships among variables and reveals the patterns in the data set. Also, ARM calculates probability and association of one variable with others and observes frequently occurring patterns. In this study, association analysis was performed by using the Apriori algorithm. This algorithm calculates and compares the frequency of co-occurrence and probability between variables in a data set [11]. As a result of the algorithm, support, confidence and lift values are obtained. Support is the total frequency of occurrence of a variable in the association analysis in the data set. Confidence shows the reliability of the relationships in the association analysis and the leverage indicates the trustworthiness of the associations. In this study, the first 10 rules (top 10) sorted by lift values and produced above minimum 10% support and minimum 80% confidence levels were used. Association analysis was performed by R Studio program and Colored Bayesian Networks were produced by JASP program. 

## 3. Results

### Characteristics of the Study Population

In total, 1118 participants completed the study and 75% were women (Table 1). The total number of participants who experienced headaches was 998, and migraine was diagnosed in 984 (88%), in which 620 (63%) experienced episodic migraine and 364 (36%) chronic migraine (CM). Among the participants, 344 (30.7%) were diagnosed with MOH. Table 1 summarizes the characteristics of the study population. Headache duration was >10 years in 379 participants, 5–10 years in 247 participants, 1–5 years in 292 and <1 year in 80 individuals. 

Typical visual aura was present in 127 (11.3%) participants, and all were diagnosed with migraine. Among them, 23 (2%) were diagnosed with MOH. The presence of aura was not significantly associated with IBS (*p* > 0.05). MOH frequency was significantly higher in patients with migraine without aura (37%) compared to patients with migraine with aura (22%, *p* < 0.001).

Overused medication was predominantly NSAIDs (*n* = 307, 89%) in MOH patients and remaining patients were abusing triptans (*n* = 37, 11%). A total of 38 out of 984 participants in the survey were using migraine prophylaxis treatment. Among them, 21 individuals were using serotonin–norepinephrine reuptake inhibitors (SNRIs), 13 individuals were on beta-blockers, 3 individuals were on a combination of SNRIs and beta-blockers, and one individual were on topiramate. 

Gastrointestinal symptoms fulfilling the IBS diagnosis were selected by 560 (50%) of the participants. Among the participants with migraine, 512 (52%) had IBS symptoms. IBS symptoms were defined by 270 (44.3%) patients with episodic migraine (EM), 17 (85%) CM patients without MOH and 225 (65%) patients with MOH and CM. In contrast, only 48 (35%) of participants without headache reported IBS. 

Migraine sufferers demonstrated significant avoidance behavior (Figure 2) towards certain food items, including packaged juices (*p* = 0.022), pasta (*p* = 0.029), canned foods (*p* = 0.035), mustard (*p* = 0.037), ready-to-eat frozen meats/chicken products (*p* = 0.003), chicken/meat bouillon (*p* = 0.009), artificially sweetened snacks (*p* = 0.037) and grapefruit (*p* = 0.014). Participants with MOH exhibited avoidance behavior towards certain food items such as fish (*p* = 0.03), soda (*p* = 0.006), milk (*p* < 0.001), orange juice (*p* = 0.005), cabbage (*p* = 0.006), lentils (*p* = 0.009), pickles (*p* < 0.001) and cream (*p* = 0.003). MOH and migraine groups avoided consuming ketchup and soy sauce, and MOH and IBS groups avoided consuming nuts (*p* < 0.001), citrus fruits (*p* < 0.001), eggplant (*p* = 0.001), bell pepper, beans and pomegranate (*p* = 0.001). All three groups avoided consuming bulgur wheat and its derivatives (*p* < 0.001) (Figure 2) (Table 2). Food triggers for migraine patients with aura were not significantly different from migraine patients without aura (*p* > 0.05) except strawberry (*p* = 0.034).

MOH patients had specific food triggers, including banana (*p* = 0.001), apple (*p* = 0.014), cherry (*p* < 0.001), apricot (*p* = 0.001), watermelon (*p* = 0.021), olives (*p* = 0.014), ice cream and yogurt (*p* < 0.001). Migraine specific food triggers included peanut butter (*p* = 0.001), chocolate, cacao (*p* < 0.001), coffee cream (*p* = 0.041), milk powder (*p* = 0.003), fish (*p* < 0.001), red wine (*p* = 0.005), white wine (*p* = 0.015), mayonnaise (*p* = 0.011), baklava, kunefe (*p* = 0.002), gateau (*p* = 0.002) and foods containing red food dye (*p* = 0.001). Both MOH and IBS were often triggered by tomato juice (*p* = 0.002) and spinach (*p* = 0.005). Cream cheese (*p* = 0.017), custard, milk (*p* < 0.001), potato, onion, garlic, bulgur wheat (*p* < 0.001), bagel, citrus fruits (*p* < 0.001), cabbage, canned foods, soy sauce and beer (*p* < 0.001) were identified as common triggers for migraine, MOH and IBS (Figure 3).

Mig-SCog scores, which assess subjective cognitive complaints during migraine attacks, were compared among groups. MigSCog scores were higher in patients with MOH (10.5 ± 5.2) compared to patients without MOH (7.20 ± 4.98) (*p* < 0.001). MigSCog was significantly higher in EM patients with IBS (8.6 ± 4.36) compared to EM patients without IBS (6.34 ± 4.03) (*p* = 0.16) and MOH patients with IBS (11.44 ± 4.17) compared to MOH patients without IBS (8.65 ± 4.4). Presence of IBS in CM patients did not make a significant impact on MigSCog scores (CM patients with IBS (10.7 ± 3.4) vs. CM patients without IBS (9 ± 3.6), *p* > 0.05). The highest score was observed in patients with MOH and IBS (Figure 4).

High carbohydrate intake and large amounts of food in a meal was also identified as a trigger in participant with migraine. Finally, chewing duration of each bite (<20 s) was also assessed, and no significant differences were found among groups.

## 4. Discussion

We investigated the co-occurrence of MOH and IBS and the food triggers for MOH, migraine and IBS. Our pioneer study showed that the frequency of IBS symptoms was higher in patients with MOH compared to patients without MOH. We found the frequency of IBS symptoms in 50% of individuals with migraine and 65% in those with MOH. A higher prevalence of IBS symptoms in patients with migraine has been reported, but the association of MOH with IBS is investigated for the first time.

IBS is the most common functional bowel disorder, presenting with symptoms such as abdominal pain, bloating and distention, along with changes in bowel movements. These changes may manifest diarrhea, constipation or a combination of both, without any apparent structural or biochemical abnormalities to explain them. The Rome criteria provide a standardized approach to define and diagnose IBS based on specific patterns of symptoms related to gastrointestinal symptoms [19]. The prevalence of IBS in the previous studies differs between 11 and 31% [20]. However, a recent study showed that IBS prevalence is around 42–45% in the population if the diagnosis is made by primary physician or gastroenterologist. Moreover, up to 75% of the patients in the with IBS population may remain underdiagnosed despite being fulfill the diagnostic criteria [4]. We found the frequency of IBS symptoms to be 50% in our general study population, 35.8% in non-headache sufferers, 52% in migraine patients and 65% in MOH patients. IBS symptoms are higher in MOH patients compared to other groups.

Despite its common occurrence, IBS is one of most underdiagnosed problems and the underlying mechanisms of IBS remain unknown. Suggested potential risk factors encompass heredity, dietary patterns, imbalances in gut microbiota, gastrointestinal infections and psychological aspects, all of which can impact the bidirectional brain–gut axis. IBS has been shown to be -associated with intestinal permeability change called leaky gut. Some studies have shown a correlation between usage of analgesic drugs and irritable bowel syndrome [21,22]. Research demonstrated that NSAIDs in physiological doses were toxic to human intestinal epithelium by interfering with mitochondrial energy metabolism and inducing oxidative stress [23]. Also, in the same study, NSAIDs were shown to facilitate the transport of intestinal compounds from luminal side to the basolateral side and damage tight junctions. The integrity of intestinal barrier function is crucial to keep lipopolysaccharide and other pathogenic molecules away from the blood circulation. Recently, we have shown in an MOH model that chronic NSAID exposure was associated with elevated serum lipopolysaccharide binding protein, occludin and vascular endothelial cadherin levels which were consistent with leaky gut [24].

In the literature, the mechanisms by which food triggers induce migraine attacks are not fully understood. Some of the investigated mechanisms include (1) the release of biogenic amines such as tyramine, which leads to nitric oxide release due to the presence of sulfur compounds found in foods, (2) the presence of histaminergic compounds in foods, and (3) changes in the levels neurotransmitters such as dopamine due to flavonoid content in foods. Some substances in migraine triggers inhibit the enzyme group ‘sulfotransferase’ which plays a role in neurotransmitter metabolism. Quercetin and catechin in chocolate, hesperidin in oranges and lemons, quinic and caffeic acid in coffee, epicatechin in tea and red wine cause inhibition of sulfotransferases (SULTs), SULT1A1 and SULT1A3 enzymes [12]. SULT1A1 plays a role, especially in the metabolism of catecholamines such as dopamine. Migraine triggers can initiate a migraine attack by inhibiting SULT1A enzymes involved in the deactivation of neurotransmitters. It has been suggested that many NSAIDs also inhibit SULT1A1 enzymes, and a recent study has shown decreased brain SULT1A1 activity in MOH model using mefenamic acid, and NSAID [25].

We have identified banana, apple, cherry, apricot, watermelon, olives, ice cream, and yogurt as food triggers specific to MOH. Additionally, patients with MOH avoided the consumption of fish, sugary carbonated beverages (e.g., soda), milk, heavy cream, orange juice, lentils, cabbage and pickles, which appeared to be a specific dietary behavior for MOH patients. Food triggers in MOH are commonly consumed healthy and natural fruits in daily life (e.g., banana, apple, cherry, apricot, watermelon). Additionally, these foods can be classified as dopaminergic foods as they are rich in dopamine content and capable of stimulating dopamine release. On the other hand, food triggers specific to migraine without MOH, are predominantly consisted of histaminergic and processed foods with additives. For individuals with IBS, red grapes, which are also dopaminergic, can act as a specific food trigger. 

It is important to identify the food triggers or be aware of the avoidance behavior for certain foods since they may indicate underlying susceptibility to migraine, MOH or IBS. Accordingly, a recent meta-analysis demonstrated that most of the patients with migraine avoid alcohol consumption probably due to the fear of a migraine attack [26]. In our study, beer and rakı were identified as common triggers of migraine, MOH and IBS, whereas red wine and white wine were reported as triggers by only migraine patients. 

Non-medical management of MOH must contain avoidance/limitation of dopaminergic foods during the discontinuation of overused analgesics. The symptoms related to leaky gut and IBS should be recognized by headache specialists and questioned for the appropriate treatment of headaches in patients. 

Migraine is a painful and debilitating neurological disease and the first leading cause of years lived with disability under age 50 [27]. Cognitive dysfunction is one of the major factors that contributes to disability in all ages [28]. Mig-SCog scores are used to assess subjective cognitive complaints during migraine attacks. Patients with MOH had significantly higher scores indicating cognitive dysfunction compared to patients with episodic migraine. Also, IBS also seemed to be a factor affecting cognitive impairment in both patients with MOH and patients with EM. Mig-SCog scores showed that the presence of comorbid IBS significantly interfered with cognitive function in migraine patients. IBS and MOH both should be taken into consideration in patients with migraine related cognitive dysfunction. 

Our study has some limitations for consideration. The recruitment of participants by an online survey does not reflect the general population, as participants who use the internet and have their own e-mail addresses and are familiar with online surveys were included. The higher rate of university graduates among participants displays that bias. Another limitation is the use of self-reported data. However, there is no substantial reason not to use self-reported data as all the characteristic features of migraine, IBS and MOH required for the diagnosis were included in the questionnaire. 

This study addresses a novel link between MOH and IBS by reaching a high number of individuals by online survey. Our study adds to the existing literature as (1) it evaluates diverse group of food items routinely consumed in daily life that are relevant to cultural habits, and (2) it assesses cognitive impairment in IBS and MOH for the first time. Lastly, we increased awareness related to food triggers, MOH and IBS via online survey.

## 5. Conclusions

The present study demonstrated that there is an association of IBS and MOH. The frequency of IBS in patients with MOH was higher than migraine patients or non-headache sufferers. MOH specific triggers were mostly dopaminergic foods, whereas migraine specific food triggers were mostly histaminergic and processed foods. IBS and cognitive complaints are seen at a higher rate in MOH patients. IBS contributes to cognitive impairment as coexistence of IBS seems to be a factor worsening cognitive functions in patients with EM and MOH. Personalized diets focusing on food triggers and interference with leaky gut must be integrated for MOH and migraine treatment to achieve sustainable management of these disorders.

## Figures and Tables

**Figure 1 jcm-12-06488-f001:**
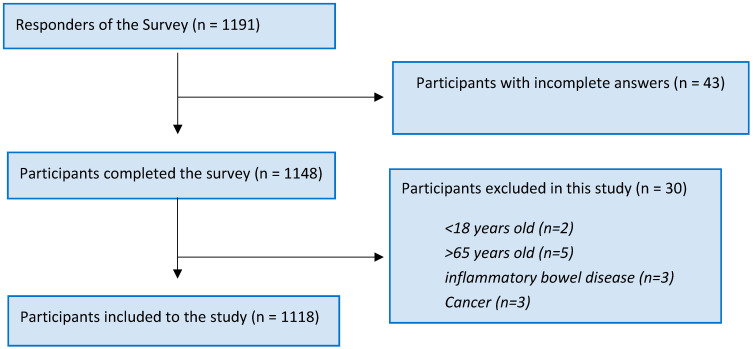
Flow chart of the study design.

**Figure 2 jcm-12-06488-f002:**
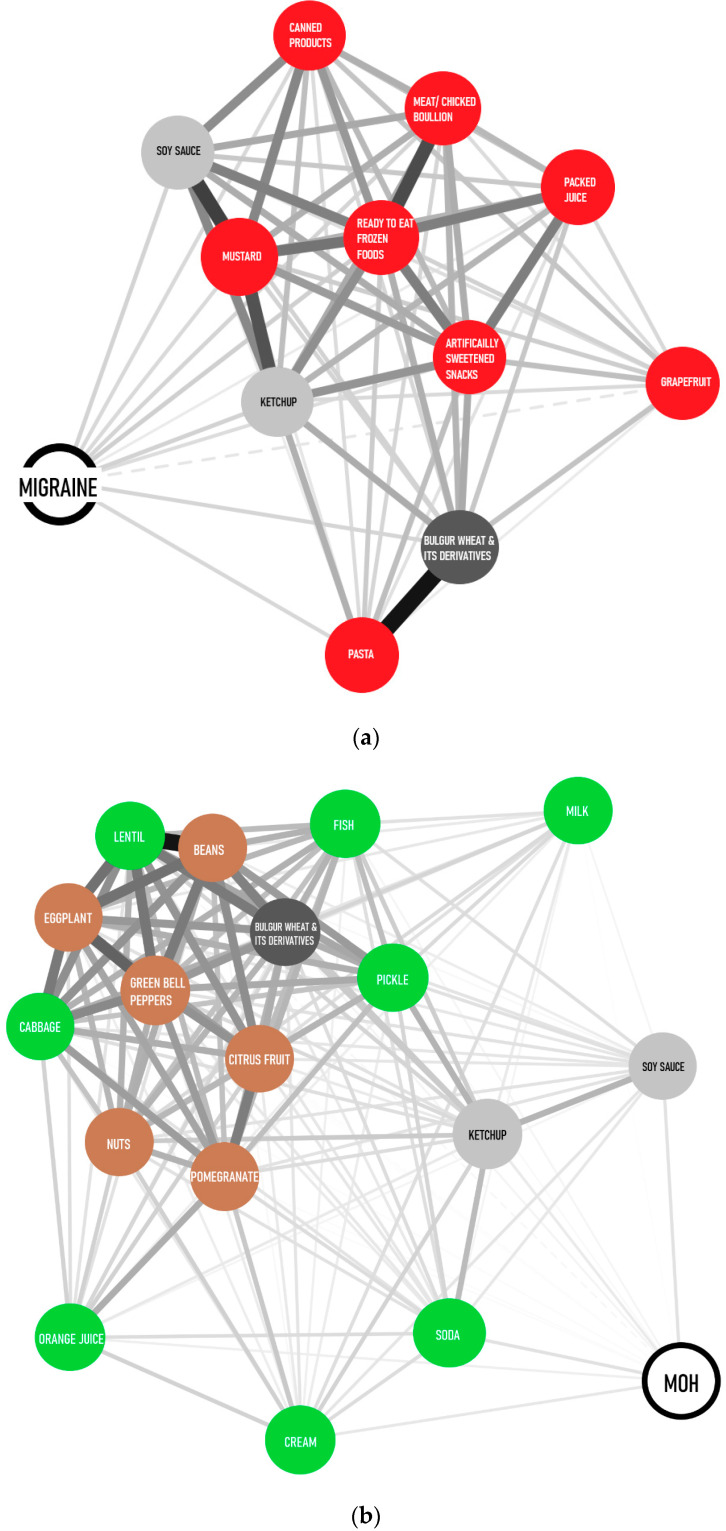
Foods avoided by the subjects. Network graphics were made according to the correlation levels among variables. Avoided foods for (**a**) migraine (**b**) MOH and (**c**) IBS are given in the figure. Groups represent structures with similar nodes clustered. Avoided foods shared by subjects with MOH, migraine or IBS are in neutral colors, and specific foods for MOH (green), migraine (red) and IBS (yellow) are in bright colors. Line thickness denotes the strength of association between two nodes in the network graphic. Cluster distance signifies the similarity or difference between groups. Nodes with similar features are closer to each other than nodes with different properties. Network Graphic is illustrated using JASP 0.14.1.0 program.

**Figure 3 jcm-12-06488-f003:**
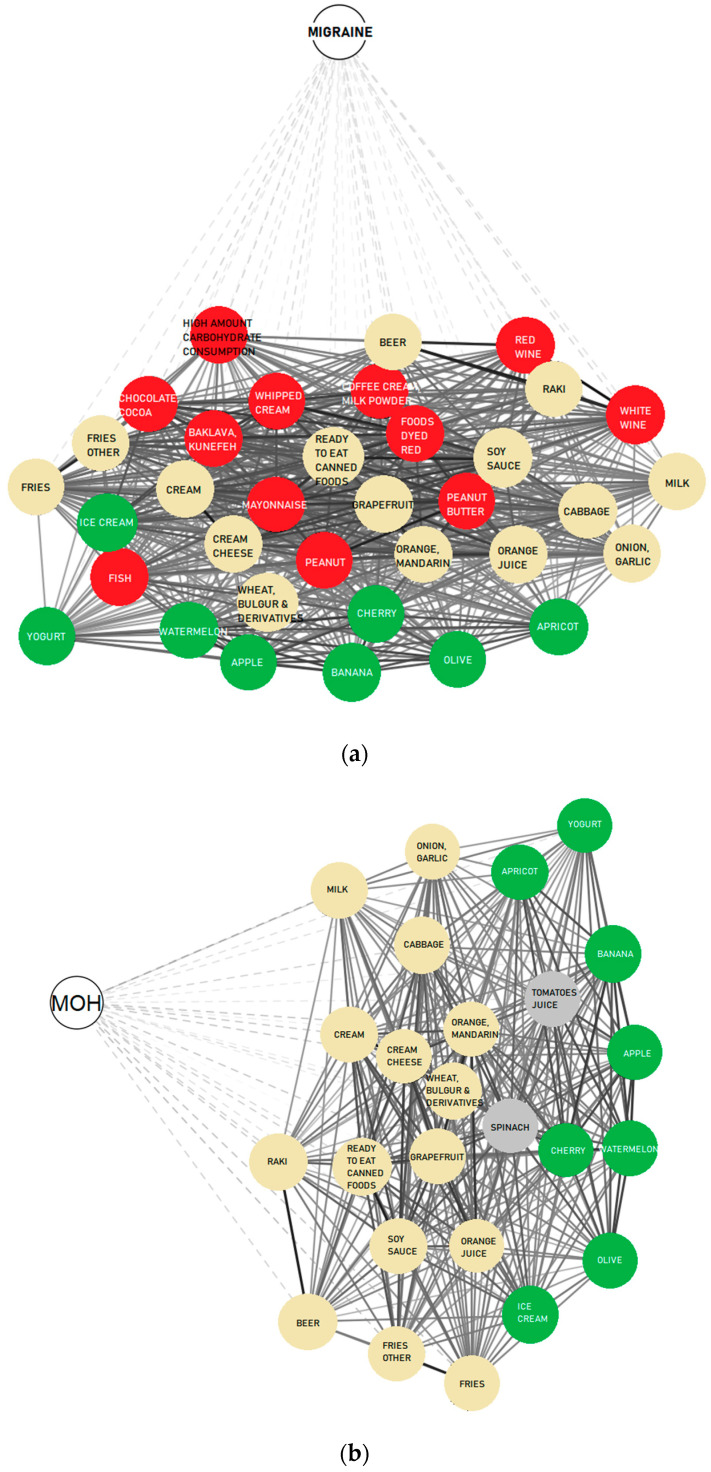
Food triggers of the subjects. Network graphics were made according to the correlation levels among variables. Food triggers for (**a**) migraine, (**b**) MOH and (**c**) IBS are given in the figure. Groups represent structures with similar nodes clustered. Food triggers shared by subjects with MOH, migraine or IBS are in neutral colors, and specific foods for MOH (green), migraine (red) and IBS (yellow) are in bright colors. Line thickness denotes the strength of association between two nodes in the network graphic. Cluster distance signifies the similarity or difference between groups. Nodes with similar features are closer to each other than nodes with different properties. Network Graphic is illustrated by using JASP 0.14.1.0 program.

**Figure 4 jcm-12-06488-f004:**
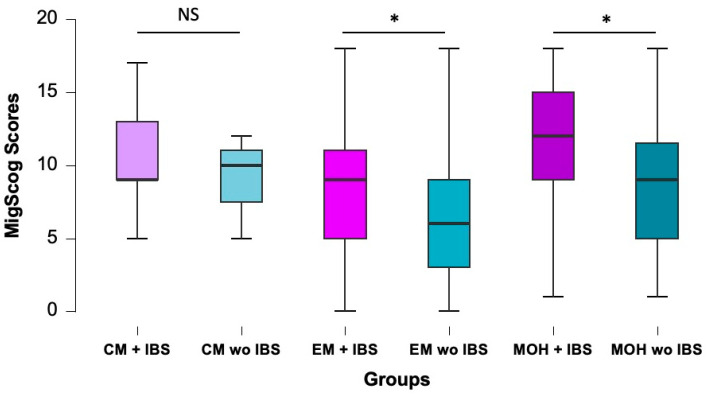
MigSCog scores of MOH and migraine patients with or without IBS. Co-existence of IBS is significantly associated with higher cognitive dysfunction scores in EM and MOH patients, *: *p* < 0.01. MigSCog, migraine attack subjective cognitive impairments scale; CM, Chronic migraine; EM, Episodic migraine; MOH, Medication overuse headache; IBS, Irritable bowel syndrome; NS: Not significant).

**Table 1 jcm-12-06488-t001:** The characteristics of the study population (*n* = 1118).

	Migraine	MOH	IBS	Participants without Headache
*n* (%)	984 (88.0%)	344 (30.7%)	560 (50.2%)	134 (12.0%)
Age (years), mean ± SD	33.96 ± 11.1	35.01 ± 11.1	32.92 ± 11.1	33.20 ± 12.5
Woman, *n* (%)	731 (75.0%)	293 (80.7%)	436 (77.9%)	62 (51.7%)
Education/University graduate, *n* (%)	267 (73.6%)	757 (76.9%)	407 (72.7%)	86 (71.7%)
Body Mass Index, mean ± SD	24.38 ± 5.8	24.15 ± 4.1	24.47 ± 6.8	24.28 ± 7.6

**Table 2 jcm-12-06488-t002:** Phytochemical properties of foods identified as triggers or avoided by MOH and migraine patients.

Foods	Food Content	Main Related Neurotransmitter	Flavonoids	SULT Inhibition	Additives **
Bakery (bread, simit, etc.)	Wheat Yeast Egg Gluten	Dopamine Histamine	Apigenin	SULT1A1	In packaged breads (Caramels, sorbic acid, propionic acid, phosphoric acid, acetic acid, stearol -2-lactylates, tartaric acid of mono and diglycerides of fatty acids esters, lactic acid)
Milk and milk products (Cheese, yoghurt, heavy cream, etc.)	Casein, whey Tyrosine Lactose Vitamins (A, B, D) Minerals (Ca, Mg, P…)	Dopamine Histamine	-	-	Carrageenan, Polyphosphates, Potassium sorbate, trisodium sorbate, calcium gluconate, sodium citrates, succinic acid, sucrose esters of fatty acids
Olives [12]	Fatty acid Oleic acid Linoleic acid Synapic acid Vanillic acid Cyanidin-3 rutinosid	Dopamine	Apigenin Luteolin Olueropein	SULT1A1 SULT1A3	-
Apple	Chlorogenic acid Pectin Ursolic acid Anthocyanin	Dopamine	Quercetin Kaempferol	SULT1A1	-
Apricot	Beta- carotene Carotenoids Ferulic acid Beta-sitosterol	Dopamine	Quercetin Kaempferol	SULT1A1	-
Banana	Caffeic acid Chlorogenic acid Carotenoids	Dopamine	Quercetin Kaempferol	SULT1A1 SULT1A3	-
Watermelon	Lycopene Cucurbitacin Citrulline Carotenoids Lutein	Dopamine	Luteolin Kaempferol	SULT1A1	-
Citrus	Ascorbic acid Limonoids Carotenoids Lutein Tangerine	Dopamine	Hesperidin Naringin	SULT1A1 SULT1A3	-
Cherry	Caffeic acid Chlorogenic acid Anthocyanin Melatonin Ascorbic acid	Dopamine	Quercetin	SULT1A1	-
Strawberry [13]	Ellagitannins Anthocyanin	Dopamine Histamine	Quercetin Kaempferol Catechin Ficetin	SULT1A1	-
Grapefruit	Limonen Naringenin Lycopene Vitamins (A, B9, C…)	Dopamine	Hesperidin Quercetin Apigenin	SULT1A1 SULT1A3	-
Red grapes	Ellagic acid Vitamins (C, K, B6, B9…) Minerals (Magnesium, Potassium…)	Dopamine	Quercetin Kaempferol Catechin Resveratrol Apigenin	SULT1A1 SULT1A3	-
Pomegranate	Punicalagin Ellagic acid Anthocyanin Flavonoids Vitamins (C, K…) Minerals( Potassium, Calcium, Iron…)	Dopamine	Quercetin Kaempferol Myricetin	SULT1A1	-
Tomatoes	Lycopene Carotenoids Malic acid Citric acid	Dopamine Histamine	Quercetin Kaempferol	SULT1A1	-
Onion/Garlic	Sulfur, organosulfur compounds Saponins Allicin	Dopamine	Quercetin Kaempferol	SULT1A1 SULT1A3	-
Cabbage	Glucosinolates Sulphoraphane Indol- 3 carbinol Vitamins (vitamin C, B9…) Minerals (Calcium, Iron, Magnesium…)	Dopamine	Quercetin Kaempferol Apigenin	SULT1A1	-
Spinach	Zeaxantine Carotenoids Vitamins (K, C, B9…)	Histamine Dopamine	Luteolin	SULT1A1	-
Eggplant	Anthocyanin Chlorogenic acid Nicotine	Histamine Dopamine	Nasunin Quercetin Lutein Rutin	SULT1A1	-
Lentils	Phytoesterols Lignans	Dopamine Histamine	Quercetin Kaempferol Myricetin	SULT1A1	-
Nuts	Eflvanoids Proanthocyanidins Phytosterols Omega-3 fatty acids Alfa tocopherol	Histamine Dopamine	Quercetin Kaempferol	SULT1A1	-
Canned foods	Acetic acid salts	It varies according to the type of food			Benzoic acid Sulfur dioxide Bisphenol A (BPA)- in tin cans
French fries [14]	Carotenoids Chlorogenic acid Caffeic acid		Quercetin Kaempferol	SULT1A1	May contain sodium bisulfite, nitrite, nitrate if it is frozen
Ketchup	Lycopene Carotenoids Malic acid Citric acid Lactic acid	Dopamine Histamine			Benzoic acid, tartrazine, modified cornstarch Xanthan gum Guar gum
Mayonnaise	Egg Yoghurt Garlic Olive oil	Dopamine Histamine			Sodium benzoate, benzoic acid Potassium sorbate Ethylene diamine tetra acetates (EDTA) Sulfur dioxide Carrageenan Colorants Modified cornstarch Sorbic acid Xanthan gum
Soy sauce	Soybean (Phytosterol, flavonoids, saponin) Wheat Salt Vinegar	Dopamine Histamine	Apigenin Luteolin	SULT1A1	Monosodium glutamate (MSG) Citric acid
Mustard [15]	Mustard seed (glucosinolate, sinigrin, Sulphoraphane, isothiocyanates, phenolic acids, flavonoids) Salt Vinegar High fructose syrup	Dopamine Histamine	Quercetin Kaempferol Catechin Myricetin		Sodium benzoate, sodium metabisulfite Carrageenan Colorants Modified cornstarch Sorbic acid Xanthan gum
Beer [16]	Barley (Flavonoids, Polyphenols, Saponarin, Kutonarin, Superoxide dismutase (SOD), GABA, Tryptophan) Malt Gluten Yeast Ethanol	Dopamine	Quercetin Kaempferol Myricetin	SULT1A1	Carbon dioxide Congener Benzoic acid, benzoates Sorbic acid, sorbates Sulfur dioxide Lactic acid Caramels Propane-1-2- diol alginate
Red wine [12]	Red grape Ethanol Yeast Tannin	Dopamine Histamine	Epicatechin gallate Resveratrol	SULT1A1 SULT1A3	Carbon dioxide Congener Benzoic acid, benzoates Sorbic acid, sorbates Phosphoric acid Sucrose esters of fatty acids Synthetic colorants
White wine	White grape [17] Ethanol Yeast Tannin	Dopamine Histamine	Naringenin Anthocyanin	SULT1A1	Carbon dioxide Benzoic acid, benzoates Sorbic acid, sorbates Phosphoric acid Sucrose esters of fatty acids Synthetic colorants
Rakı [18]	Pimpinella anisum Ethanol Tannin	Dopamine	Quercetin Luteolin Izoretientin Izovitexin	SULT1A1	Sulphite
Chocolate/Cocoa	Cocoa mass Egg products Gluten Sugar Butter oil (from milk) Palm oil	Dopamine	Quercetin Vanillin	SULT1A1	Soy lecithin Polyglycerol polyricinoleate
Coffee	Caffeic acid Quinic acid Chlorogenic acid	Dopamine	Trigonelline (coffee beans)	SULT1A1 SULT1A3	Dipotassium phosphate Sodium polyphosphate Silicon dioxide Pentasodiumtriphosphate Acacia gum Sodium aluminum silicate
Carmosine [12] (Red food dye)		Dopamine		SULT1A1	Synthetic colorant, E122
Artificially sweetened snacks	It may contain palm oil, gluten, cocoa, cream, sugar depending on the product.				Tartrazine, Brilliant blue FCF, carmosine, Green S Aspartame Sucralose Neohesperidin C
Meat/Chicken Bouillon	Salt, sugar, beef fat, dried chicken meat, chicken fat, parsley, soybean oil, whey, sodium caseinate				Monosodium glutamate (MSG), Corn starch, hydrolyzed corn protein, autolyzed yeast extract, disodium inosinate, disodium guanylate, citric acid, onion powder, yellow 5, yellow 6, Anotta
Packaged juice	Apple, apricot, cherry, watermelon, etc. Sodium salt Potassium salt High fructose syrup	Dopamine Histamine	Hesperidin Quercetin Kaempferol	SULT1A1 SULT1A3	Acesulfame KA Acesulfame K Aspartame Saccharin Sucralose Carboxy methyl cellulose Sorbic acid, sorbates

**: The food additives have been written based on the Turkish Food Codex.

## Data Availability

Not applicable.

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
