# Peer review of "Distinct Food Triggers for Migraine, Medication Overuse Headache and Irritable Bowel Syndrome"

_jcm, 2023, doi:10.3390/jcm12206488_

Round 1
Reviewer 1 Report
Dear Special Issue Editor Marija Veslinovic, I have no more comments to the paper. It was clear. The study was elegant. The minor weaknesses are already mentioned in the paper! I guess that the group of patients with MOH primarily are patients with tension type headache. Thus, it fits in the ordinary picture of tension related disorders. However, it also fits in the condition chronic multicanalicular BPPV (mc-BPPV) in which tension type headache is a common symptom, in 80 - 90 percent of the patients. IBS in patients with mc-BPPV is also common, and is explained via the vagus nerve.: In patients with mc-BPPV abnormal signals are transmitted from a diseased labyrinth to the healthy functioning vestibular nuclei complex in the brainstem. From there, the labyrinthine abnormal signals go directly via the vestibulo-spinal reflexes or via different cranial nerve nuclei to their end organs, which consequently act in an abnormal way. Vestibular migraine is explained via the vestibulo-thalamocortical reflex.I hope the authors find this interesting and will consider a similar study.
Congratulations with a very fine and interesting study! Well performed!
Author Response
Dear Editor in Chief Journal of Clinical Medicine,
We have revised the manuscript according to the reviewers’ comments and changes are highlighted in the text. Our responses are below;
Reviewer 1
We thank the reviewer 1 for her/his valuable comments and suggestions.

Reviewer 2 Report
The paper presented to me for review deals with a very practically relevant issue, which is the effect of food triggers on headaches. The topic is so far little understood despite many years of clinical observations. The authors of the paper undertook to investigate IBS frequency in medication overuse headache patients and identify food triggers and food avoidance behavior.
The study is based on a very large group of patients (n=1118). The methodology is based on a questionnaire and was described in detail and transparently. In the course of their study, the authors showed that the frequency of IBS was higher in MOH patients compared to migraine patients a coexistence of IBS seems to be a confounding factor for cognitive functions.
The paper is written in good language, the figures and tables are clear and undoubtedly bring new knowledge on the topic. However, before accepting the paper for publication, I would supplement the work with a few aspects that would increase its merit:
1. a rather controversial issue is alcohol as a trigger because the latest huge meta-analysis, which should be included in the discussion: PMID: 37612595 - showed that it is not exactly a relevant trigger and most often patients avoid alcohol consumption out of fear of a migraine attack
2. what criteria were used to diagnose the type of headache? ICHD-3? IHS? this should be completed
Author Response
Dear Editor in Chief Journal of Clinical Medicine,
We have revised the manuscript according to the reviewers’ comments and changes are highlighted in the text. Our responses are below;
Reviewer 2
We thank the reviewer 1 for her/his constructive comments.
Question 1:
a rather controversial issue is alcohol as a trigger because the latest huge meta-analysis, which should be included in the discussion: PMID: 37612595 - showed that it is not exactly a relevant trigger and most often patients avoid alcohol consumption out of fear of a migraine attack
Suggested reference and related sentence was added to the discussion section.
Question 2:
what criteria were used to diagnose the type of headache? ICHD-3? IHS? this should be completed.
‘All headaches were diagnosed according to ICHD-3.’ The sentence was added to the methods section.
